# Structure Effect on the Response of ZnGa_2_O_4_ Gas Sensor for Nitric Oxide Applications

**DOI:** 10.3390/nano12213759

**Published:** 2022-10-26

**Authors:** Ray-Hua Horng, Shu-Hsien Lin, Dun-Ru Hung, Po-Hsiang Chao, Pin-Kuei Fu, Cheng-Hsu Chen, Yi-Che Chen, Jhih-Hong Shao, Chiung-Yi Huang, Fu-Gow Tarntair, Po-Liang Liu, Ching-Lien Hsiao

**Affiliations:** 1Institute of Electronics, National Yang Ming Chiao Tung University, Hsinchu 30010, Taiwan; 2Graduate Institute of Precision Engineering, National Chung Hsing University, Taichung 402, Taiwan; 3Department of Post-Baccalaureate Medicine, College of Medicine, National Chung Hsing University, Taichung 402010, Taiwan; 4Integrated Care Center of Interstitial Lung Disease, Taichung Veterans General Hospital, Taichung 407219, Taiwan; 5Department of Critical Care Medicine, Taichung Veterans General Hospital, Taichung 407219, Taiwan; 6Thin Film Physics Division, Department of Physics, Chemistry, and Biology, Linköping University, 58183 Linköping, Sweden

**Keywords:** NO gas sensor, ZnGa_2_O_4_, response, first-principles calculation

## Abstract

We fabricated a gas sensor with a wide-bandgap ZnGa_2_O_4_ (ZGO) epilayer grown on a sapphire substrate by metalorganic chemical vapor deposition. The ZGO presented (111), (222) and (333) phases demonstrated by an X-ray diffraction system. The related material characteristics were also measured by scanning electron microscopy, transmission electron microscopy and X-ray photoelectron spectroscopy. This ZGO gas sensor was used to detect nitric oxide (NO) in the parts-per-billion range. In this study, the structure effect on the response of the NO gas sensor was studied by altering the sensor dimensions. Two approaches were adopted to prove the dimension effect on the sensing mechanism. In the first approach, the sensing area of the sensors was kept constant while both channel length (L) and width (W) were varied with designed dimensions (L × W) of 60 × 200, 80 × 150, and 120 ×100 μm^2^. In the second, the dimensions of the sensing area were altered (60, 40, and 20 μm) with W kept constant. The performance of the sensors was studied with varying gas concentrations in the range of 500 ppb~10 ppm. The sensor with dimensions of 20 × 200 μm^2^ exhibited a high response of 11.647 in 10 ppm, and 1.05 in 10 ppb for NO gas. The sensor with a longer width and shorter channel length exhibited the best response. The sensing mechanism was provided to explain the above phenomena. Furthermore, the reaction between NO and the sensor surface was simulated by O exposure of the ZGO surface in air and calculated by first principles.

## 1. Introduction

It is well-known that the internet of things (IoT) can be used to connect messages received anytime and anywhere through a large database and to transmit data in real time [1,2,3,4,5,6]. Various sensors can be applied to receive different types of information as used in IoT. A gas-sensor-incorporated smart-home control system can reduce air pollution, improve security, and save energy [7]. Nowadays, there are many gas sensors on the market, such as electrochemical, optical, capacitance-based, calorimetric, and metal oxide-based types. Each of these can be used to detect variation in the concentration of toxic gases in order to maintain a safe environment and avert accidents. Among the above-mentioned gas sensors, the dimensions of metal oxide-based gas sensors are the smallest, making them suitable for commercial applications in environmental monitoring and healthcare. Furthermore, semiconductor metal oxide based gas sensors can be fabricated at low cost; they have low power consumption and a long lifetime, and they possess the ability to operate in harsh environments [8,9].

Many studies have shown that the type and concentration of the gas exhaled by a patient can be used for preliminary diagnosis [10,11,12]. Different types of gas sensors can be utilized to identify the presence of toxic gases in the environment [13,14,15]. Among the different gas sensors, those for the detection of nitric oxide (NO) have received more attention because the gas exhaled by a human contains more than 35 ppb of NO, which indicates a high possibility of asthma [16,17,18,19]. According to the 2019 Global Asthma Report, about 262 million people worldwide suffer from asthma [20]. If NO gas sensors can be used accurately and non-invasively to monitor human breath, then they will change the current biomedical detection ecology. Thus, the NO gas sensor capable of detecting low parts-per-billion concentration has considerable research value.

The sensing material ZnGa_2_O_4_ (ZGO) is a special spinel structure that provides special surface dangling bonds, and it only reacts with NO because of its surface-bonding-energy characteristics [21]. There are dangling bonds of Zn and Ga on the surface, which have excellent adsorption capacity for NO. As for the NO-sensing mechanism of the ZGO gas sensor, it has been published in our previous work [21]. The high selectivity between NO and NO_2_, CO, CO_2_, and SO_2_ has also been demonstrated [21]. The gas sensor with parallel plates as the electrode configuration was used [22]. The resistance of the film increases when molecular NO is adsorbed. This means that this characteristic, which in turn changes the conductivity of the film, can be used to sense NO gas concentration. Additionally, the response can reach the required parts-per-billion level. Because of the excellent response to NO and low-level detection of the ZGO sensor, it can be applied not only for the monitoring of exhaust gas emissions from automobiles but also for monitoring the breathing of patients with asthma [22].

Although the ZGO epilayer has been demonstrated to be a good NO gas sensor, the sensing structure has not been discussed for optimizing the sensing performance. The mechanism depends on the electrical resistance variation resulting from NO gas concentration. In order to optimize the gas sensor performance, different dimensions of sensors were studied in this work.

## 2. Experimental

A ZGO thin film was grown on a c-plane sapphire substrate, Al_2_O_3_(0001), by metalorganic chemical vapor deposition (MOCVD). Diethylzinc (DEZn) and triethylgallium (TEGa) were used as the precursors of Zn and Ga, respectively. Purified oxygen (O_2_) and argon (Ar) were employed as the oxygen source and carrier gas, respectively. The growth temperature was set at 600 °C. The thickness of the as-grown film was estimated to be approximately 100 nm under a growth rate of 0.8 nm/min. After the film growth, annealing treatment at 700 °C in an ambient air was performed for 60 min to further improve the film quality. The material characteristics of the ZGO film were analyzed by X-ray diffraction (XRD), scanning electron microscopy (SEM), transmission electron microscopy (TEM), energy-dispersive X-ray spectroscopy (EDS) and X-ray photoelectron spectroscopy (XPS). The active area of the device was defined through patterning the ZGO film and created by a dry etching process using a high-density inductively coupled plasma reactive-ion etching system. Afterward, the metal electrodes titanium (Ti)/aluminum (Al)/titanium (Ti)/gold (Au) with 20 nm/300 nm/20 nm/100 nm, respectively, were deposited on the ZGO layer using an electron-gun evaporation system. The metals can directly play the Ohmic contact with the ZGO epilayer. It was not necessary to do any thermal annealing. The schematic diagram of the gas sensor is shown in Figure 1. 

After fabrication of the gas sensors, a thermal stage was used to heat the device to ensure their response characteristics. The measuring setup with 8 L volume has been described in our earlier work [21]. The operating temperature was 300 °C. Various concentrations of NO gas were injected to evaluate the response of the gas sensor. For the response measurement of the gas sensor, it was not necessary to offer the gas flow rate continuously. The known concentration of gas was just injected into the chamber and diluted by the air in the close chamber. In our system, the mass flow controller was 1 sccm. If the concentration of NO was higher 0.5 ppm, the concentration of NO with 4% was controlled by the injection duration. This means that a 2 min injection of 4% NO gas can create a 10 ppm NO concentration and a 6 sec injection of 4% NO can create a 0.5 ppm NO concentration. Concerning the ppb range of NO concentration, the 80 ppm NO gas and syringe were used. The 20 cc syringe with 80 ppm NO gas was injected into the 8 L camber and could create a 200 ppb NO concentration. The 1 cc syringe with 80 ppm NO concentration injected into the chamber could create a 10 ppb NO concentration. The electrodes of the sensor devices were connected to an Agilent B1505A to measure the change in resistance of the sensing films. After measurement, the chamber gas was vacuumed by pump and air was injected to 1 atm for the next cycle measurement.

## 3. Results and Discussion

Before studying the structure’s effect on the response of the gas sensor, the material properties of the as-grown film were characterized by XRD, SEM, TEM, EDS, and XPS. As shown in Figure 2a, the 2θ-scan XRD pattern reveals only three main peaks located at 18.63, 37.71, and 58.01^o^ indexed corresponding to the (111), (222), and (333) of ZGO, respectively, and one peak from Al_2_O_3_(0006) located at 41.7° [21]. The result indicates that the film might be grown in a highly-oriented single crystal spinel ZnGa_2_O_4_ (ZGO) structure. The inset of Figure 2a shows the surface morphology of the ZGO film measured by SEM. Apparently, the ZGO film consisted of nanospindles with a dimension in length and width of about 120 nm and 40 nm, respectively. Such a nanospindle structure gives rise to an increasing surface-to-volume ratio, which would enhance the performance of the gas sensor. To ensure the formation of the spinel ZGO structure, the microstructure and elemental distribution of the film were further investigated by TEM and EDS. Figure 2b shows a bright-field TEM image and a corresponding EDS line scan across from the surface towards the substrate. The uneven surface reflects the nature of the formed nanospindle structure. The elemental line profiles can be divided into three distinct regions: (i) surface coated Pt film; (ii) nanospindle film composed of Ga, Zn, and O; and (iii) sapphire substrate composed of Al and O. The intensity of elemental profiles evolutes with the same trend, revealing a thickness dependence, and indicating a uniform compositional distribution without obvious phase separation throughout the entire film. The microstructure of the film characterized by high-resolution TEM, shown in Figure 2c, clearly presents atomic-resolved crystalline lattices despite forming a granular structure. Figure 2d shows a Fourier-filtered lattice-plane image taken from the area marked with an orange rectangle in Figure 2c. Although some misfit dislocations were formed between grains, the lattice is still resolved. The interplanar spacing of the lattice was calculated to be 4.816 Å, which matches well with the interplanar spacing of ZGO(111). In combination with the corresponding fast Fourier transform pattern, shown in the inset of Figure 2d, we confirm that the film was grown in a spinel ZGO structure, which agrees well with the XRD result.

Furthermore, XPS of ZGO film was measured and used to evaluate the chemical composition of ZGO film. All peaks were calibrated with the binding energy of 284.8 eV of *C 1s* peak. The software casaXPS was used for data analysis. Figure 3a shows that there exist two peaks with binding energy of 1021.78 and 1044.78 eV corresponding to the Zn 2p orbital. The binding energies of Ga 2p located at 1117.78 and 1144.68 eV were also observed, as shown in Figure 3b. These obtained binding energies match well with their corresponding oxide compounds, indicating that Zn and Ga atoms bond to oxygen atoms. [23,24] As to the obtained oxygen spectrum O 1s, shown in Figure 3c, the peak reveals asymmetrical shape. The spectrum was further deconvoluted using Gaussian-Lorentzian (Voigt) peak shape and Shirley-type background. The fitting yields three main peaks, which can be assigned as O_I_ (530.58 eV), O_II_ (531.08 eV) and O_III_ (532.18 eV) related to lattice oxygen, oxygen atoms in the vicinity of an oxygen vacancy and chemisorbed water, respectively [25]. It was found that there were 35 % and 8 % for the oxygen vacancy and chemisorbed water by surface dangling bonds relative the O 1s total amount. Because the gas sensor was operated at 300 °C, the O_II_ and O_III_ would contribute to react with NO gas. The O_III_ could be desorbed, which produce more dangling bonds and contribute to increasing the response of NO gas sensor.

To understand the performance of gas sensors with a constant sensing area, sensors varying in both length and width were fabricated. The dimensions of the sensing area, length (L) × width (W), for the fabrication of devices were 60 × 200, 80 × 150, and 120 × 100 μm^2^. In each case, the area was maintained at 12,000 μm^2^. The schematic diagrams of the sensing devices are shown in Figure 4. Channel length is defined as the distance between the two sensing electrodes, that is, the distance through which the electron travels in the ZGO film. On the other hand, channel width is defined as the length of the electrode perpendicular to the channel. This determines the cross-sectional area through which the electrons can pass.

In order to evaluate the effect of sensing area on the NO response, five different concentrations of NO gas (10, 5, 2.5, 1, and 0.5 ppm) were injected into these gas sensors. The operating temperature was 300 °C. Note that the response can be obtained as Rg/Ra. Rg is the resistance of the sensor in the analyzed gas and Ra is Resistance of the sensor in dry air. Figure 5 shows the change in resistance with exposure to NO gas. Figure 5a shows the response diagram of the 60 × 200 μm^2^ gas sensor. As the concentration decreases, the response also decreases. The responsivities of the five different NO concentrations to the thin film were 5.086, 2.489, 1.936, 1.638, and 1.422. Note that the based resistance of the sensor was about 90 kΩ. Figure 5b shows the response of the 80 × 150 μm^2^ thin film sensor. The responses to different NO gas concentrations were 4.003, 2.463, 1.856, 1.741, and 1.306. The based resistance of the sensor was 230 kΩ. Figure 5c shows the response of the 120 × 100 μm^2^ film. The responses to different NO gas concentrations were 1.794, 1.792, 1.549, 1.453, and 1.261, and the highest resistance recorded by the sensor was 550 kΩ. The obtained response for all sensors is shown in Table 1.

In general, the based resistance is given by Equation (1):R = *ρ**L*/*WT*,(1)
where *ρ* is the resistivity of ZGO; *L* and *W* are the length and width of the sensing area, respectively; and *T* is the thickness of the epilayer. As all sensors use the same epilayer having the same resistivity and thickness, the relation of resistance for these sensors is R (120 μm × 100 μm) > R (80 μm × 150 μm) > R (60 μm × 200 μm). The relation is consistent with the obtained resistance. Furthermore, the gas sensor with the shorter length (60 μm) presented the highest response among these gas sensors. This is because the surface of the small channel length easily absorbs the NO and the depleting electrons are close to the surface. It requires more NO gas to deplete the surface electrons for the wider length. A more detailed response mechanism will be discussed later.

After the fixed-sensing-area test, a sensor with a constant channel width of 200 μm was tested. The samples with different channel lengths of 60, 40, and 20 μm are shown in the inset of Figure 6. NO concentrations of 10, 5, 2.5, 1, and 0.5 ppm were used to examine the response of the sensors. Figure 6 shows the transient response of the resistance of three different sensors with a constant width. Figure 6a shows the responses to five different NO gas concentrations of the films with an area of 60 × 200 μm^2^, which were 5.086, 2.489, 1.936, 1.638, and 1.242. The based resistance recorded by the sensor was 90 kΩ. Figure 6b is the response diagram of the sensor with 40 × 200 μm^2^ size; the responses of the sensor were 7.956, 5.448, 3.319, 1.770, and 1.497, respectively. The based resistance recorded by the sensor was 50 kΩ. Figure 6c presents the sensor with a 20 × 200 μm^2^ area; the responses were 11.647, 8.633, 3.469, 2.453, and 2.105, respectively. The based resistance recorded by the sensor was 9 kΩ.

Figure 7 shows the response of the sensors with different values of L as a function of NO concentration. The gas concentration was varied from 0.5 to 10 ppm. At the same electrode width, the shorter channel length exhibited better responses at each concentration. This is consistent with the phenomenon discussed earlier in which, under the same area, a shorter channel length results in greater responsiveness. Although the sensing area is different for the sensors shown in Figure 5, as the channel length becomes shorter, the response increases. It is important to mention that the 20 μm channel length presented the best response among all gas sensors. Nevertheless, the linearity of a gas sensor of 20 μm × 200 μm was not good in the range. It is necessary to undertake further study on the mechanism of non-linearity. In order to evaluate the limitation of the gas sensor with the 20 μm channel length, the inset of Figure 7 shows the response as a function of low NO gas concentration. It was found that the response was 1.05 even when the NO gas concentration was 10 ppb. The NO gas sensor with 20 μm × 200 μm sensing area exhibited linear behavior. Not only that, but the response to high concentrations (>10 ppm) of NO gas was also measured, shown in Figure 7b. The responses were 12.5 and 16 for the 25 and 50 ppm NO concentration, respectively. It also presented linear behavior.

Due to the NO gas presenting a response up to 10 ppb, it has the potential to be applied to asthma care. It is well known that the human exhaled breath contains a large amount of water. Thus, the evaluation of the effect of humidity on the response of the gas sensor is important. Here, the best performance of the gas sensor with the 20 μm × 200 μm sensing area was used to measure the relative humidity (RH) 35.3, 50.4 and 78.7%. As the NO gas concentration was 5 ppm, the response of the gas sensor operating at 300 °C was 9.228, 8.633 and 8.293 for an RH of 35.3, 50.4 and 78.7%, respectively, shown in Figure 8a. It was found that the response of the ZGO presented a 10% decay as the RH increased. The decay was not obvious because the gas sensor was operating at 300 °C. In order to clarify the effect of operation temperature on the response of the gas sensor, Figure 8b presents the response of the ZGO gas sensor as a function of NO concentration with different temperatures. Obviously, the response increases as NO concentration and operation temperature increase. As mentioned in the XPS analysis, the surface of ZGO absorbed the OH from the environment. It is necessary to desorb the OH and produce more dangling bonds, which contribute to increasing the response of the NO gas sensor.

From our previous study, the grains of ZGO nanostructures are covered with adsorbed oxygen molecules before NO injection [21]. Because of the higher electronegativity, the adsorbed oxygen molecules accept electrons from the conduction band of metal oxides and form oxygen ions (O^2−^) [26,27,28,29,30,31]. As NO is injected into the sensing chamber, the concentration of free charge carriers is reduced because of the removal of electrons from the metal oxide; therefore, a depletion layer is formed at the grain boundaries [29]. Because of the high electron affinity of NO (2.28 eV) compared with oxygen (1.46 eV), the NO molecules interact with adsorbed oxygen ions on the ZGO surface and enhance the potential barrier at grain boundaries, which further increases the resistance.

From the results, it is clear that the response of the sensor improves with the decrease in channel length. The response of the sensor is defined for the oxide gas NO and is given as:(2)$$\mathrm{Response}=\frac{R_{g}}{R_{a}}=\frac{R_{a}+\Delta R}{R_{a}}=1+\frac{\Delta R}{R_{a}},$$
where *R_g_* is the resistance after injection of the gas, *R_a_* is the resistance before gas injection, and Δ𝑅 is the increasing resistance upon gas injection. Low *R_a_* and an increase in Δ𝑅 with gas injection can enhance the response of the sensor. The resistance before gas injection is defined in Equation (3):(3)$$R_{a}=\rho \frac{L}{A}=\rho \frac{L}{W\times T}\propto \frac{L}{W},$$
where *ρ*, *L*, *W*, and *T* are the resistivity, channel length, channel width, and thickness of the film, respectively. Since *ρ* and *T* are constant for the present work, *R_a_* depends only on *L* and *W*. Equation (3) shows that the resistance before injecting gas is directly proportional to the length of the sensor and inversely proportional to the width. For the first case, in which the sensors have the same area, *R_a_* of the sensor with 60 μm × 200 μm showed the lowest base resistance. With regard to the NO injection, the shortest channel presented the highest response. Furthermore, at the same *W*, the 20 μm length presented the smallest base resistance. Concerning the resistance variation after injection of the NO gas, exposure to Zn and Ga dangling bonds causes NO to turn into absorbing NO^−^. These NO^−^ ions are replete with surface electrons. As the channel was shorter, the depletion region became deeper. This resulted in a big change in resistance. The reaction mechanism is shown in Figure 9. For the same channel width, the resistance changed significantly after injection of NO gas. This resulted in the high response of the ZNO gas sensor.

Further considering the charges transfer between nitric oxide and ZGO, this can be calculated using the Δ𝑅. For example, the gas sensor was operated at 2V and the base resistance was 9 kW for the gas sensor with a sensor with dimensions of 60 μm × 200 μm. This means the base current was 0.22 mA (2 V/9 kΩ). The 1 ppm concentration of NO gas reacted with the ZGO gas sensor, which resulted in the resistor change to 18 kΩ (shown in Figure 6c), indicating the sensing current was 0.11 mA. The charge transfer can be calculated to have 6.8 × 10^14^ electrons (0.11 mA/1.6 × 10^−19^) from ZGO to NO.

As discussed above, the ZGO gas sensor with various sensing areas was changed to measure the response to NO. Before sensing NO, the ZGO surface attracts oxygen molecules in the air to form oxygen ions, and the surface oxygen ions act as catalysts to react with the NO molecules. The reaction between NO and the sensor surface has received insufficient attention. The oxide passivation of ZGO epilayers in air is simulated by O exposure of the ZGO surface to air. We performed first−principles calculations to study the mechanism of NO reaction on the oxide-passivation ZGO surface in detail, and calculated using the Vienna ab initio simulation package (VASP) code [32,33,34]. The calculations were performed using the Perdew–Wang generalized gradient approximation (GGA) for the exchange-correlation energy [35,36]. The electronic configurations for valence electrons are N 2*s*^2^2*p*^3^, O 2*s*^2^2*p*^4^, Zn 3*d*^10^4*p*^2^, and Ga 4*s*^2^4*p*^1^. The ZGO slab chosen for this study is based on previously reported structures of the Ga–Zn–O-terminated ZGO (111) models [21,37,38]. The cell parameters of the ground-state ZGO slab are *a* = *b* = 11.85 Å, *c* = 33.48 Å, α = *β* = 90°, and *γ* = 120°, which are sufficient to decouple the interactions between NO or NO_2_ molecules, as well as top and bottom interactions. We consider four plausible models showing an NO molecule approaching O exposure on the ZGO surface as depicted in Figure 10. The reference energy corresponds to the energy of the O exposure on the ZGO surface and a free NO molecule as shown in Figure 10a. In Figure 10b, a free NO_2_ molecule forms and consists of a free NO molecule bound to a surface oxygen ion away from the ZGO surface; in Figure 10c,d, an NO molecule bonds to a surface oxygen ion on the ZGO surface to form an NO_2_-like molecule on the ZGO surface. According to our previous study, the nitrogen atom from a single NO_2_ molecule to the Ga atom on the Ga-Zn-O-terminated ZnGa_2_O_4_ (111) surface exhibits the highest work function change of +0.97 eV [38]. Here, we adopt a Ga atom on the ZnGa_2_O_4_ (111) surface as the position of the initial adsorbed site. The stoichiometry of the four surface models was fixed at Zn_16_Ga_32_O_72_N_1_O_1_. The atomic coordinates of each surface model were fully relaxed to their zero force positions. A plane wave cutoff of 450 eV was used in conjunction with a 3 × 3 × 1 gamma-centered grid to achieve a force accuracy of 1.0 × 10^−4^ eV/atom. Our results show a free NO bound to a surface oxygen ion to form a free NO_2_ molecule away from the ZGO surface with an energy gain of −1.79 eV from a reference state, as shown in Figure 10b. When adsorbed on O-on-ZGO films, a NO molecule reacts with a surface oxygen ion to form an NO_2_-like molecule with a bond angle of 131.12° and two bond lengths of 1.24 Å, as well as an energy gain of −2.65 eV from a reference state, as shown in Figure 10c. The bonding of a nitrogen atom from a NO_2_-like molecule to a Ga atom on the ZGO (111) surface with an equilibrium bond length of 2.52 Å is thermodynamically favorable under the conditions described in our growth experiments, further validating the results of NO oxidized to NO_2_. This shows that ZGO presents high selectivity for NO and NO_2_ [21]. In Figure 10d a nitrogen atom from an NO molecule is adsorbed on a surface oxygen ion on the ZGO surface to form an NO_2_-like molecule with a bond angle of 138.10° and two bond lengths of 1.18 and 1.20 Å, as well as an energy gain of −2.44 eV from a reference state. The characteristics of this NO_2_-like molecule are closer to a free NO_2_ molecule with a bond angle of 136.4° and two bond lengths of 1.20 Å. Finally, regardless of NO or NO_2_, Figure 10 shows that NO or NO_2_-like molecules cause an increase in the work function change or resistance response, which has been demonstrated in our past experiments and in the current study [21,37,38].

The gas adsorption-induced electrical resistance changes are intimately related to their respective electronic bonding properties. The total charge densities of the oxide-passivation ZGO surface, the oxide-passivation ZGO surface with one surface oxygen removed, and a nitrogen (oxygen) atom from a NO_2_-like molecule to Ga atoms on the ZGO (111) surface are shown in Figure 11. The contour levels are drawn at intervals of 0.038 electrons Å^−3^, with the lowest contour level at 0.038 electrons Å^−3^. Before adsorption of nitric oxide, the charge densities around the oxide-passivation ZGO surface are 0.038 electrons Å^−3^ irrespective of the presence of surface oxygen being removed as observed in Figure 11a,b. After nitric oxide adsorption, we clearly see that nitric oxide and surface oxygen tend to combine significantly to form NO_2_ with smooth isosurfaces and the charge density as high as 0.190 electrons Å^−3^ is localized on the midpoint positions between NO_2_ and surface gallium atoms as shown in Figure 11c. This suggests that the adsorption of nitric oxide on the surface of ZGO confines the electrons near the adsorption site, which reduces the conductivity and increases the resistance. In the case of a nitrogen atom from an NO molecule bonded to a surface oxygen atom to form a NO_2_-like molecule with smooth isosurfaces, electrons with a charge density of 0.152 electrons Å^−3^ occupy the midpoint positions between NO_2_ and surface gallium atoms as shown in Figure 11d.

To further elucidate the gas response behavior in our slabs, in Figure 12 we have plotted the electronic band structures of the oxide-passivation ZGO surface, the oxide-passivation ZGO surface with one surface oxygen removed, and a nitrogen (oxygen) atom from an NO_2_-like molecule to Ga atoms on the ZGO (111) surface. For the first Brillouin zone, we used Γ-point at the center of the zone. The B-points at (2π/*a*)(1/2,0,0) are the intersections with the zone surface along the six equivalent <100> directions. In addition, the F-points at (2π/*a*) (0,1/2,0) are the intersections with the zone surface along the six equivalent <010> directions, where *a* is the lattice constant. The band structures of the oxide-passivation ZGO surface and the oxide-passivation ZGO surface with one surface oxygen removed are shown in Figure 12a,b, which represent direct band gaps of 2.73 eV and 2.77 eV, respectively, which closely match the other theoretical gap of the ZGO bulk of 2.82 eV and severely underestimate the experimental band gap of 5.18 eV for the ZGO bulk due to the well-known shortcomings of the strong hybridization of inner electrons [39,40,41]. Note also that the appearance of a defect level in the forbidden bandgap is caused by the removal of one surface oxygen atom. Furthermore, when a nitrogen atom from an NO_2_-like molecule is adsorbed on the surface of ZGO (111), the band structure shows a significant increase in the band gap up to 3.02 eV as shown in Figure 12c, and this fact should be taken into account in interpreting the high response of the gas sensor to NO gas. In contrast, when an oxygen atom from an NO_2_-like molecule is adsorbed on the surface of ZGO (111), the band gap is slightly increased to 2.79 eV as shown in Figure 12d, suggesting a less favorable response of the gas sensor to NO gas.

## 4. Conclusions

We have demonstrated a gas sensor with an excellent response to NO gas, which was achieved with the wide-bandgap material ZGO grown by MOCVD. By studying the effect of the device’s dimensions on the sensing behavior, we found that a shorter channel length and longer channel width of the gas sensor resulted in a greater response. The best pattern size of the gas sensor was 20 (L) × 200 (W) μm^2^, and the response of this pattern could reach 2.105 in 500 ppb of NO gas. Furthermore, the gas sensor presented a high response of 1.05 in 10 ppb of NO gas. The above results indicate that the ZGO epitaxial layer has a high potential for NO gas sensor applications. A sensing mechanism is also reported for NO adsorption on O-on-ZGO films via surface oxygen ions as catalysts to form NO_2_-like molecules for achieving the desired ultra-high response.

## Figures and Tables

**Figure 1 nanomaterials-12-03759-f001:**
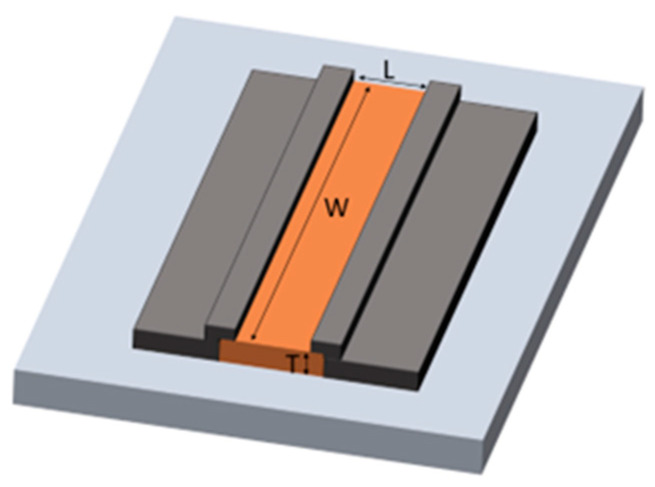
Schematic diagram of the ZGO gas sensor.

**Figure 2 nanomaterials-12-03759-f002:**
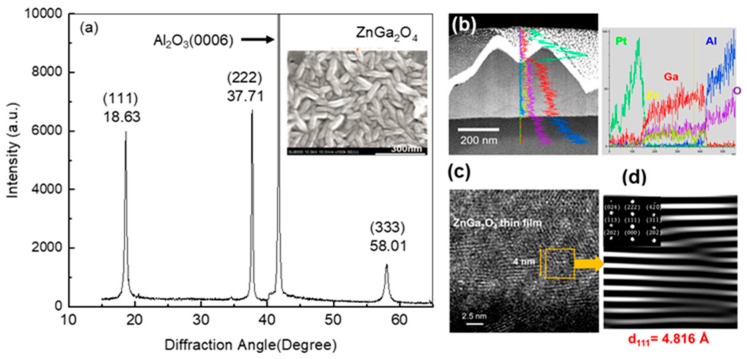
Material characteristics of the as-grown ZGO film. (**a**) θ/2θ-scan XRD pattern and corresponding top-view SEM micrograph, shown in the inset, of the film. (**b**) Microstructure and corresponding elemental distribution measured by TEM and EDS. (**c**) Lattice-resolved TEM image taken from the film, and (**d**) Fourier-filtered lattice-plane image converted from the area marked with an orange rectangular in (**c**), as well as a corresponding fast Fourier transform pattern shown in the inset of (**d**).

**Figure 3 nanomaterials-12-03759-f003:**
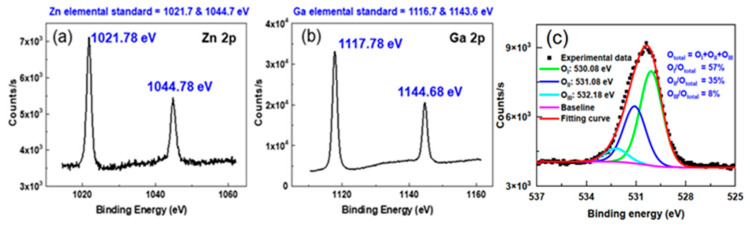
(**a**) Zn 2p, (**b**) Ga 2p and (**c**) O 1s XPS spectra of the ZGO thin film.

**Figure 4 nanomaterials-12-03759-f004:**
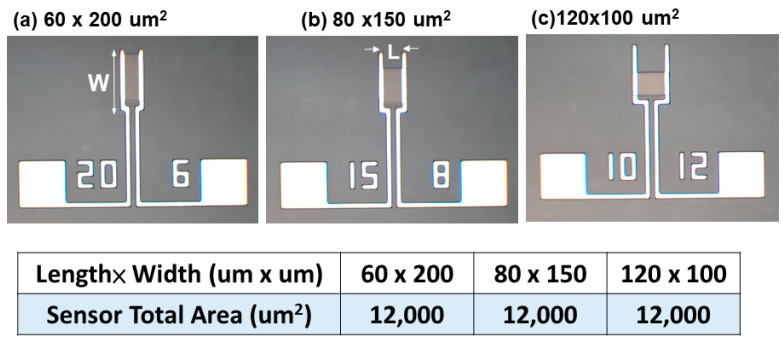
Structures of the devices with a fixed area of 12,000 μm^2^. The dimensions of the sensing area, L × W, are (**a**) 60 × 200 μm^2^, (**b**) 80 × 150 μm^2^, and (**c**) 120 × 100 μm^2^.

**Figure 5 nanomaterials-12-03759-f005:**
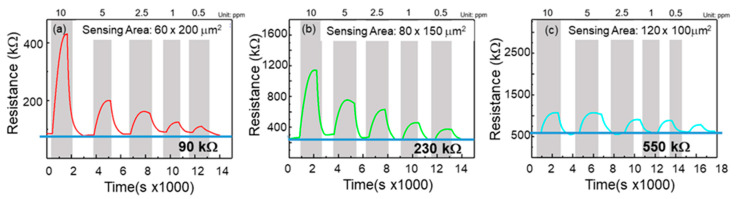
Resistance as a function of injection gas concentration for the gas sensors with a fixed sensing area but with different dimensions (L × W) of (**a**) 60 × 200 μm^2^, (**b**) 80 × 150 μm^2^, and (**c**) 120 × 100 μm^2^.

**Figure 6 nanomaterials-12-03759-f006:**
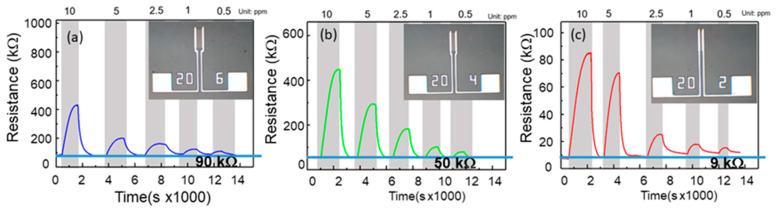
Resistance as a function of injection gas concentration for the gas sensors with a fixed channel width of 200 μm and varying lengths of (**a**) 60 μm, (**b**) 40 μm, and (**c**) 20 μm. Insets show the structures of the devices.

**Figure 7 nanomaterials-12-03759-f007:**
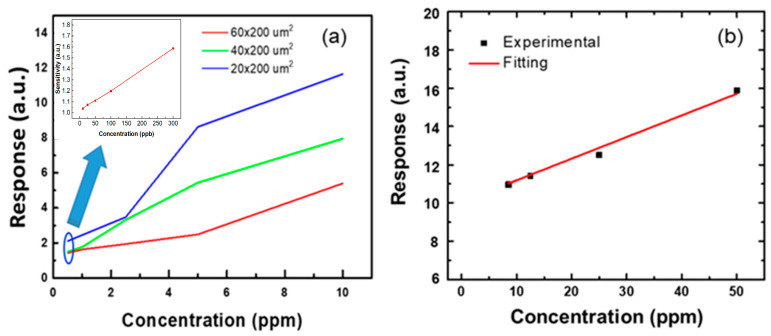
Response of the gas sensor devices as function of (**a**) low NO gas concentration for the gas sensor with different length and (**b**) of high NO gas concentration. Inset of Figure 6a shows the response of ZGO with 20 μm × 200 μm as a function of NO gas with ppb concentration.

**Figure 8 nanomaterials-12-03759-f008:**
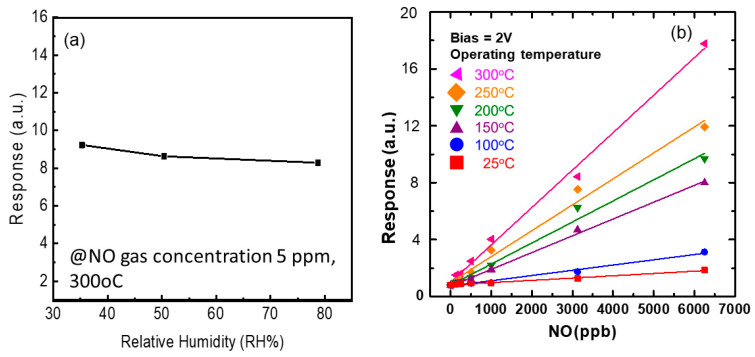
Response of ZGO gas sensor (**a**) as function of relative humidity measured at 5 ppm of NO concentration and (**b**) as function of NO concentration operated at different temperature.

**Figure 9 nanomaterials-12-03759-f009:**
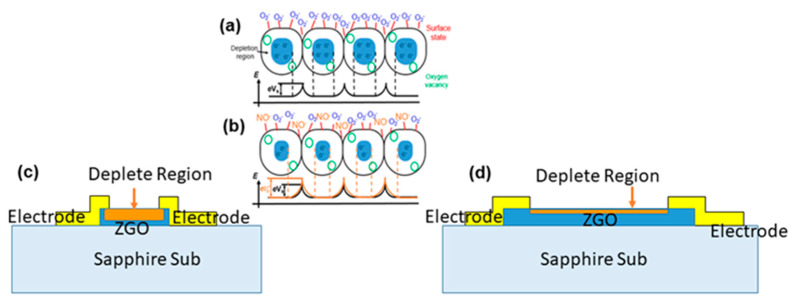
(**a**) Surface states of the gas sensor before NO gas injection and (**b**) electron depletion due to the absorption of NO gas during NO gas injection. Resistance variation mechanism for (**c**) short and (**d**) long−channel−length gas sensors.

**Figure 10 nanomaterials-12-03759-f010:**
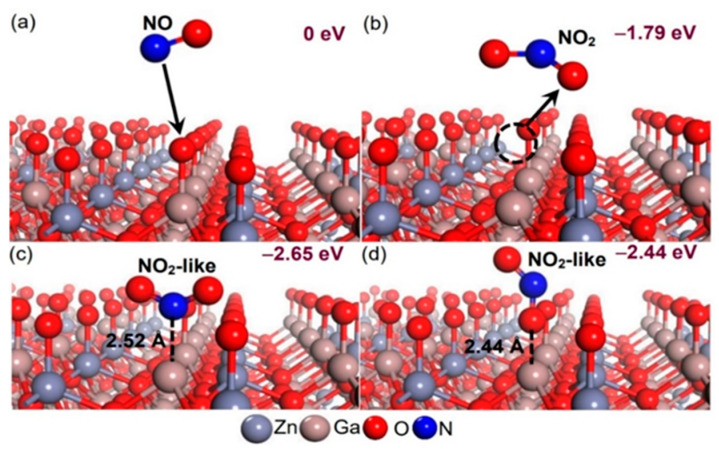
Sequence shows the (**a**) NO exposure on the oxide-passivation ZGO surface, (**b**) NO_2_ away from the ZGO surface, (**c**,**d**) NO_2_-like molecules on the oxide-passivation ZGO surface. The atoms are represented by spheres: Zn (gray, large), Ga (brown, large), O (red, medium), and N (blue, medium).

**Figure 11 nanomaterials-12-03759-f011:**
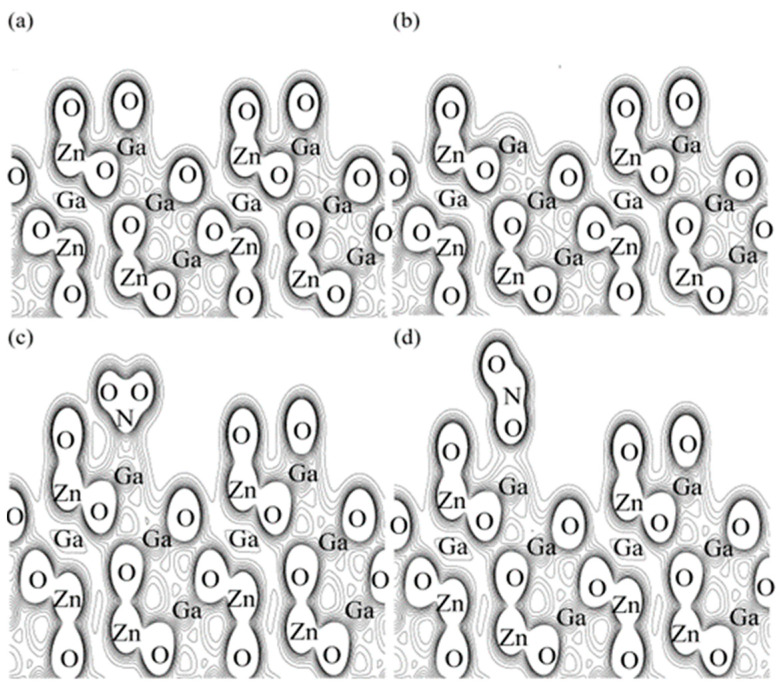
Valence charge density maps on (**a**) the oxide-passivation ZGO surface, (**b**) the oxide-passivation ZGO surface with one surface oxygen removed, (**c**,**d**) NO_2_-like molecules on the oxide-passivation ZGO surface. The charge is normalized to 36 electrons per unit slab. The contours are plotted in a range from 0.038 to 0.418 electrons Å^−3^ in 10 steps.

**Figure 12 nanomaterials-12-03759-f012:**
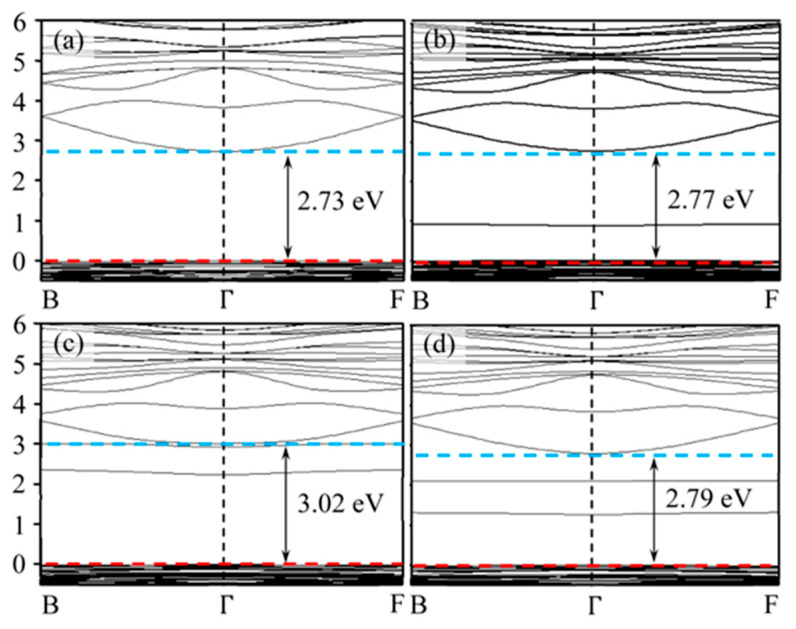
GGA band gaps of (**a**) the oxide-passivation ZGO surface, (**b**) the oxide-passivation ZGO surface with one surface oxygen removed, (**c**,**d**) NO_2_-like molecules on the oxide-passivation ZGO surface. The blue and red dashed lines represent the conduction band minimum and valence band maximum, respectively.

**Table 1 nanomaterials-12-03759-t001:** Responses of the gas sensors with different gas injections.

	Gas Concentration (ppm) and Corresponding Response				
Sensor Dimension (μm^2^)	10 (ppm)	5 (ppm)	2.5 (ppm)	1 (ppm)	0.5 (ppm)
60 × 200	5.086	2.489	1.936	1.638	1.422
80 × 150	4.003	2.463	1.856	1.741	1.306
120 × 100	1.794	1.792	1.549	1.453	1.261

## Data Availability

Data sharing is not applicable to this article.

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
