# Peer review of "Structure Effect on the Response of ZnGa2O4 Gas Sensor for Nitric Oxide Applications"

_nanomaterials, 2022, doi:10.3390/nano12213759_

Round 1
Reviewer 1 Report
(1) The authors need to provide details parameters that were used for their measurements. For example, there is no information on what is the gas flow rate that was used during this experiment.
(2) I am not sure the statement of "single phase and highly oriented single crystal" is accurate for this thin film. The authors need to revise it.
(3) I don't think the current XPS results is sufficient to assess the quality of ZGO film. The XPS measurements were done to assess the chemical state of each element in the film. If the purpose is to examine properly the chemical composition, they should also provide wide scan results in this manuscript.
(4) The XPS analysis is not appropriate and properly done. The authors should compare the obtained binding energies with that of element in similar state. For example, if you want to analyze Zn2+ as oxide, you should compare it with the binding energy of Zn2+ as oxide compound and not with with Zn as metal state. What the authors did was comparing binding energy for Zn in oxide with metal standard, this is not apple to apple comparison. Please re-do the analysis for XPS results.
(5) The authors also need to reveal what is the software and what parameter that was used for fitting analysis of all XPS peaks. Also why only O2p that was fitted? It is strange that all peaks have no background/baseline? Why O2p is deconvoluted into 3 peaks? The O2 peak width does not seem that wide to me.
(6) It is common to observe OH group in O2 spectra when examining surface, especially when the surface was not etched prior to measurement. However, the XPS measurement seems to be done at RT while the measurement of sensor characteristic were done at 300oC. What makes the authors sure that OH groups are still exist or stable on the surface of the thin film at 300oC?
(7) The authors should mention in the manuscript that the sensor response is R/Ro. It would also be better if the authors also provide separate or additional figures for response in Fig. 5 & 6 as "R/Ro vs Concentration", I know the authors provided small explanation in the text, but having it presented in figures would make it easier to understand.
(8) I don't know why the authors put figure 5 and 6 in different scale. They are better be in the same scale.
(9) Fig.7, why there is a change in linearity trend when the concentration is higher than 5 ppm? Also, for blue line, there seems to be 2 linearity trend, why is that?
(10) What does "a little decay" in line 232 means?
(11) Caption in Fig. 8, it says "humility", but I think the authors meant "humidity".
(12) How can the authors conclude that the sensor response is repeatable when the authors only did 2 sets of measurements? It should be measured 10 times at minimum to determine that this response is repeatable or not. What is "original" means in Fig. 9(b)?
(13) why all sensors responses value are positive?NO2 is oxidizing gas while the rest of the tested gases is reducing gases, why all gases gave the same positive responses?
(14) Line 290: " For the same channel width, the resistance changed significantly after injection of NO gas". Is this phenomenon only occur for NO? How about other gases?
(15) As stated in line 357, upon adsorption, NO will form NO2 on the surface. The authors' calculation predict that such process will occur at Ga site. Why such process is not likely to occur at Zn site?
Author Response
Please refer the attached file.

Reviewer 2 Report
In the submitted manuscript, the Authors have proposed a very interesting, well organized and argued paper.
Also, the proposed application is attractive, and the used material is promising.
In my opinion the paper is supported by a thorough characterization of the ZGO film and by a detailed description of the sensing mechanism. The geometry of the device is well optimized, and the results are encouraging.
However, the fabrication description (row 89-91) should be more clear.
Furthermore, the Authors should pay attention to the use of the word “Responsivity”: it is not the same of “response of the sensor” and can be confusing. It is used typically in photodetector systems, while the correct definition of “Response” is well reported by the Authors in formula (2) and is perfect to define the sensors behaviour.
Regarding the Fig. 8(b), is not clear if the Authors would show the connection between the humidity and the temperature, why they performed measurements under humidity only for a temperature and for one gas concentration? Please clarify.
Finally, to be taken as a suggestion for future developments, why the Authors did not think to an interdigitate electrode (IDT) structures?
Author Response
Please refer the attached file.

Round 2
Reviewer 1 Report
There are a lot of my comments that are being ignored by the authors.
(1) My request to put more information on experimental details is not only pertaining to the gas flow. Gas flow is only one example that I put out. The authors need to provide detail parameters on how they run their experiments.
(2) I cannot understand this sentence "it is not necessary to offer gas flow rate continuously". What is that mean? The authors mentioned that they adjust the concentration of gas using flowmeter, they must be using flow rate to adjust the concentration. Does the total gas flow rate change every time you change the concentration?
(3) I still do not understand why the authors refuse to put the background/baseline on the XPS peaks. It is unacceptable. Please check how other professionals in XPS analyze and properly present the XPS results. As an example, please look at this paper: https://doi.org/10.1021/acsnano.2c06747
(4) My previous request that the authors should provide explanation on how the authors calculate sensor response was also being ignored. The authors must provide that information in the text.
(5)The authors mention that for gas concentration higher than 5 ppm, the sensors were measured using the different diluted gas. What is that mean? I dont think that answer my question so i am going to put them back in here again. Fig.7, why there is a change in linearity trend when the concentration is higher than 5 ppm? Also, for blue line, there seems to be 2 linearity trend, why is that?
(6) You cannot say that you measured long term stability after only 5 days measurement. That is inaccurate. The authors must revised that.
(7) I suggest the authors remove Fig. 9(a), because it is only 2 times measurements, you cannot claim that it is repeatable after only 2 measurements, plus judging from the results, they are not repeatable. So that figure has no significance.
(8) Let me rephrase my previous question 13. why both NO and NO2 generated positive response? one is reducing gas but the other is oxidizing gas.
(9) Why significant change of resistance only occur for NO but not for other gases?
(10) I strongly suggest the authors get some help from English language editor.
Author Response
Please refer the attached file.

Round 3
Reviewer 1 Report
NO is not oxidizing gas, but NO2 is reducing gas (https://www.mdpi.com/1424-8220/13/9/12467)
That statement should be revised or removed from the text.
I think I understand that the authors do not have any explanation for why responses to NO and NO2 are in the same direction.
Author Response
NO is not oxidizing gas, but NO2 is reducing gas (https://www.mdpi.com/1424-220/13/9/12467)
That statement should be revised or removed from the text.
I think I understand that the authors do not have any explanation for why responses to NO and NO2 are in the same direction.
Resp.: Before measuring the NO and NO2 response again, the Fig. 9 “Response of ZGO gas sensor for NH3, CO, CO2, NO2 and NO with 10 ppm gas concentration” was removed from the text. Please check the revised manuscript.
